# Rule-based habitat suitability modelling for the reintroduction of the grey wolf (*Canis lupus*) in Scotland

**Vashti Gwynn**[ID]*, **Elias Symeonakis**

Department of Natural Sciences, Manchester Metropolitan University, Manchester, United Kingdom

* vashtigwynn@gmail.com

**Data Availability Statement:** The dataset of random sample points used for the statistical test for difference between the models is available from the Zenodo repository (URL: https://doi.org/10.

## Abstract

Though native to Scotland, the grey wolf (*Canis lupus)* was extirpated c.250 years ago as part of a global eradication drive. The global population has recently expanded, now occupying 67% of its former range. Evidence is growing that apex predators provide a range of ecological benefits, most stemming from the reduction of overgrazing by deer–something from which Scotland suffers. In this study, we build a rule-based habitat suitability model for wolves on the Scottish mainland. From existing literature, we identify the most important variables as land cover, prey density, road density and human density, and establish thresholds of suitability for each. Fuzzy membership functions are used to assign suitability values to each variable, followed by fuzzy overlay to combine all four: a novel approach to habitat suitability modelling for terrestrial mammals. Model sensitivity is tested for land cover and prey density, as these variables constitute a knowledge gap and an incomplete dataset, respectively. The Highlands and Grampian mountains emerge strongly and consistently as the most suitable areas, largely due to high negative covariance between prey density and road/human density. Sensitivity testing reveals the models are fairly robust to changes in prey density, but less robust to changes in the scoring of land cover, with the latter altering the distribution of land mainly through the 70–100% suitability range. However, in statistical significance tests, only the least and most generous versions of the model emerge as giving significantly different results. Depending on the version of the model, a contiguous area of between 10,139km$^2$ and 18,857km$^2$ is shown to be 80 to 100% suitable. This could be sufficient to support between 50 and 94 packs of four wolves, if the average pack range size is taken to be 200km$^2$. We conclude that in terms of habitat availability, reintroduction should be feasible.

## Introduction

The grey wolf (*Canis lupus*) is native to Scotland, but was extirpated by humans c.250 years ago [1, 2]. This persecution was part of a global eradication effort that brought overall wolf numbers to their lowest point between the 1930s and 1960s [3]. However, due to subsequent legal protection and conservation, the wolf population has expanded once again, and now

5281/zenodo.6299108, DOI:10.5281/zenodo.
6299108).

**Funding:** The author(s) received no specific
funding for this work.

**Competing interests:** The authors have declared
that no competing interests exist.

occupies 67% of its former global range, including substantial expansion in mainland Europe [4]. It is, therefore, unnecessary to re-establish wolf populations in the UK in order to conserve the species, but there is growing evidence that the presence of native apex predators brings with it a range of ecological benefits [2, 4, 5]. Ripple et al. [4] showed that large carnivores are necessary to maintain biodiversity and ecosystem function, and that their roles cannot be fully reproduced by humans, and Atkins et al. [5] state that the elimination of large carnivores can suppress plant regeneration, due to population expansion and behaviour changes in herbivores. Moreover, the grey wolf can cause mesopredator cascades (affecting both mesopredators and their prey), and tri-trophic cascades (affecting every level of the food-web down to plants) [4]. Such benefits are needed in Scotland, where deer densities are beyond ecological sustainability, and where red deer can reach a density of 150/km$^2$ in some areas in winter [2, 6]. This has a serious impact on the structure, composition and function of Scottish ecosystems, especially on tree regeneration, through overgrazing and over-browsing [2, 6]. The 1995 reintroduction of grey wolves into Yellowstone National Park is considered instructive as to what may happen should wolves be reintroduced to the Scottish Highlands, as they share almost identical key species (grey wolves, elk/red deer, aspen) [2]. In Yellowstone, just a few wolves may have had profound effects, including tri-trophic cascades that ultimately improved river hydrology, and increased abundance and diversity in many species [2], although see Mech [7] for a cautionary note on this.

Nilsen et al. [1] predict that if wolves were present in Scotland for 60 years, deer densities would decline to 7/km$^2$, with >50% reduction in some places. This is in line with the Deer Commission for Scotland's target of 6/km$^2$, and would greatly relieve the current financial burden of annual hind culling in pursuit of this target [1]. Additionally, it is proposed that the re-establishment of the "Landscape of Fear" would produce behavioural changes in deer, and thus ecosystem benefits, beyond what reduction of numbers could achieve [2, 8]. Other benefits could include significant wolf-related tourism and carbon sequestration due to regenerating woodland [1, 4].

While wolf reintroduction in Britain is not currently being considered, the government's 25 Year Environment Plan sets out policy commitments to provide "opportunities for the reintroduction of native species" [9]. Reintroductions and rewilding are currently popular, and the reintroduction of another keystone species–beavers–has received much attention and support [10, 11]. There is also growing emphasis on the ecological importance of intact ecosystems, e.g. see Plumptre et al. [12]. In light of this, and the well-documented possible benefits of apex predators outlined above, the feasibility and desirability of wolf reintroduction in the UK needs to be assessed. Manning et al. [2] note the importance of a pre-existing body of research should reintroduction be considered in the future. This study is limited to Scotland because it is the area of the UK likely to be most suitable, due to its extensive deer-filled wild lands and low human density [10]. Additionally, only the Scottish mainland was considered, as this is where any reintroduction programme would likely take place. Previous studies have explored some aspects of large predator reintroduction in Scotland, including modelling hypothetical impacts of wolves on the deer population [1, 13], and mapping habitat and likely population expansion if lynx were reintroduced [14, 15]. Wilson's review [10] finds that there is likely sufficient area and prey availability in the Highlands to support a viable wolf population. Sandom et al. [13] take into account some habitat elements in their model of a hypothetical large fenced Highland reserve containing wolves. However–to our knowledge–no one has yet created a wolf habitat suitability model for all of mainland Scotland.

There are many existing predictive habitat suitability models for wolves in countries where they are already extant (notably in Italy, Switzerland, Poland, Germany and the northern USA) [16–25]. Usually, the environmental characteristics of the areas in which wolves are

already present are used to train the model (often a logistic regression model), which is then applied across the country or region to identify other areas that may be suitable. The situation in Scotland is fundamentally different, as wolves are not currently extant there, and neither Italy, Switzerland, Poland, Germany nor the northern USA can be considered sufficiently similar to Scotland, as regards land cover, climate and elevation, to be able to apply their habitat selection models directly. Fuzzy logic analysis is widely used in predictive modelling, including marine and aquatic habitat suitability modelling [26–29]. It recognises marginal locations that sit on the boundaries of classes by assigning *likelihood* of class membership to each location [26, 30]. However, with the exception of the study by Zabihi Afratakhti et al. [30] fuzzy logic analysis is notably absent from the field of terrestrial habitat suitability modelling. Thus its use in this study represents a novel approach to mammalian habitat suitability modelling.

Here, we assess habitat suitability for wolves in mainland Scotland employing a rules-based approach, based on existing knowledge about wolf ecology [26], and fuzzy logic analysis, which allows for dataset inaccuracies and uncertainty in both the definition of attribute classes and in the measurement of the phenomenon. We carry out a sensitivity analysis of the model for the variable whose suitability is most uncertain (land cover, due to a lack of data on the suitability of open habitats), and for the variable for which we have the least data (prey density). Along with the use of fuzzy logic analysis, this allows for the incorporation of uncertainty in modelling and subsequent decision-making [31]. This is particularly important in habitat suitability studies, as they contain numerous possible sources of error and/or uncertainty, e.g. spatial data inaccuracies, definition of rules based on other environments, etc [31, 32].

Habitat suitability is, of course, not the only factor to consider in any reintroduction programme. Public attitudes, and the economic, social and psychological impacts of wolf presence–especially on rural livelihoods and communities–will be key factors in any proposal to reintroduce them [10]. Though we recognise the importance of these factors, they fall outside the scope of this paper for the following reasons:

- This is intended to be a habitat suitability study, i.e. an enquiry into where, if anywhere, in Scotland meets the wolf's needs in terms of the physical characteristics of the landscape.

- Attitudes to wolves, and our economic, social and psychological relationship with them are complex and varied. The subject deserves a more extensive treatment than we could give it in a habitat suitability study.

- Attitudes, and economic, social and psychological impacts are dynamic, and opinions on rewilding and reintroduction are volatile (see Public Attitudes below). If these factors were incorporated into a habitat suitability model, that model would very quickly become outdated and therefore irrelevant.

However, in recognition of their importance, we here include an overview of farming in Scotland and public attitudes to wolf reintroduction, in order to provide context.

### Farming in Scotland

Wolves generally prefer to predate wild ungulates where available, but they can also predate livestock [10], and therefore the potential impact on Scottish farmers is an important consideration. Elsewhere in Europe, attacks occur mostly on sheep (and goats, but this is not relevant for Scotland), and occasionally on cattle or horses [10, 33]. Predation risk is notably increased at night, where livestock is free-roaming and unsupervised, and when flocks are large [10, 33, 34]. Conversely, adaptive husbandry and preventative measures can keep predation rates low,

which reduces compensation costs, but also helps build tolerance by avoiding emotional trauma, inconvenience, and negative feeling towards wolves [34].

In Scotland, only 2.5% of the population is employed in agriculture, but just over 70% of the land area is agricultural land, [35]. The majority of the farmland is not suitable for arable, with 86% falling into the Less Favoured Area category, and is used primarily for cattle and sheep farming [35]. In 2018, there were 6,600,000 sheep and 1,750,000 cows in Scotland, with many farms keeping both [35, 36]. Most sheep are grazed extensively and unsupervised outside, in both uplands and lowlands, and only brought in occasionally for lambing or extreme weather [10, 36]. Such husbandry practices would increase predation risk if large predators were ever reintroduced [10]. Sheep farming is not lucrative, and accounted for only 7% of the value of Scotland's agricultural output in 2018, despite the large land area devoted to it [35]. Indeed without subsidies (which in 2018 were >£500 million across the Scottish agricultural sector) sheep farms would run at a substantial loss, and even with subsidies 19% of Scottish farms made a loss in 2020/21 –the highest in the UK [35, 37]. Sheep output has reduced in size since 2008, and the number of holdings with sheep has also reduced [35].

A further consideration is the potential for conflict with the commercial interests of deer hunting estates. However, as mentioned earlier, such estates may in fact benefit Wolves are more likely to hunt hinds, fauns, and weak deer, while hunters are more interested in shooting large stags, so wolf predation may actually relieve estates of the considerable financial burden of hind culling [1].

## Public attitudes

Data on public attitudes over the last few decades suggests that enthusiasm for rewilding and species restoration doesn't necessarily translate into enthusiasm for wolf reintroduction. It also demonstrates how attitudes differ in different communities.

Research in the 1990s found that depending on the method of engagement, 63–86% Scottish public supported beaver reintroduction, though opposition was higher amongst angling, farming and fishing interests [10]. In 2000, 90% of the public and 64–65% of farmers and game-keepers were shown to be in favour of pine marten reintroduction in parts of England [10]. In contrast, in the early 2000s 35 out of 37 farmers in Dorset expressed negative or ambivalent views about recolonising wild boar, a species which, like wolves, can cause agricultural damage and are sometimes perceived as being dangerous [10].

In 1997, the Macaulay Land Use Research Institute carried out research in Scotland into attitudes to large carnivore reintroduction and found that 36% were in favour of wolf reintroduction, but that this dropped to 17% in Glen Affric, an area of Scotland where wolf reintroduction was being mooted at that time [10].

In 2007, Nilsen et al. [1] studied the attitudes of the rural and urban Scottish public to wolf reintroduction, using questionnaires distributed in the Glen Affric area, and in Inverness and Edinburgh. They found that on a scale between -18 to +18, urban respondents had a mean attitude score of +5.3, while rural respondents had a significantly lower but still positive score of +1.9. However, responses from farmers averaged -4.7. Deer control and tourism were the main perceived benefits, the former especially in rural areas. However, 54% rural respondents worried about danger to livestock, whereas 35% of urban pop worried about danger to people. The attitudes of people other than farmers were found to reflect media coverage of the wolf issue, which was more often positive than negative, and the attitudes of all sections of the public were found to be less extreme than those of the bodies that represented them. For example, the attitude score of the National Farmers' Union was -16 while that of the rewilding organisation Trees for Life was +18. Nilsen et al. theorised that farmers may be less opposed than

expected because the value of a sheep is low, and farm income usually comes instead from subsidies (see Farming in Scotland above), which would not be reduced by predation. In 2009, Scottish Natural Heritage stated that it did not believe there was the necessary support among the Scottish public in general and land managers in particular, and therefore it was not considering the reintroduction of wolves [38].

In recent years, many rewilding projects have been established in Scotland [39]. In 2021, the Scottish Rewilding Alliance issued a call to the Scottish Government to make Scotland the "first rewilding nation" [40]. This was matched by a motion in the Scottish Parliament calling for the same, which is currently supported by >15 MSPs [41]. The motion made reference to the high level of public support for rewilding, and a concurrent survey found that 76% of the public felt that "protecting and restoring" Scotland's nature should be a priority for the Scottish Parliament [11, 42]. A 2019 YouGov poll of just over 2000 British adults found 36% strongly supported and 46% somewhat supported reintroduction of lost native species. Of those, 44% supported reintroduction of wolves specifically, which can be calculated to 36% of respondents overall, i.e. including those that didn't support any type of reintroduction. Support was found to be higher in younger age groups, and also, interestingly, in Scotland, where overall support for wolf reintroduction can be calculated as 45% [43].

Though rather scant and disparate, these studies and polls suggest a high level of support for rewilding in general, and high support for the reintroduction of some species, but a lower level of support for wolf reintroduction. They also show the same division between the general public and those most likely to suffer losses as was observed by Wilson [10] and Nilsen et al. [1]. There are some myths influencing opinion that could be addressed, for instance, there is no evidence of non-rabid wolves ever attacking humans [44], and there are measures that would reduce risk to livestock [33, 34]. An information campaign that addressed these issues may mitigate some concerns. Nevertheless, at present wolf reintroduction lacks the public support that would be necessary for a successful release programme.

## Methods

### Study area

Scotland is a north-west European country of 78,352km$^2$, occupying the northern third of the island of Great Britain (Fig 1) [45, 46]. It has a temperate oceanic climate, that is wetter in the west with milder winters. Mean temperature in the coldest month is approximately 4˚C, and in the warmest month, 14˚C. Annual precipitation ranges from 635mm—>1000mm east to west, and significant snow falls on land above 460m in the winter [45]. Glaciated in the Pleistocene, the Highlands in the north are mountainous and rugged, whereas the Central Lowlands are relatively flat, and the Southern Uplands are hilly [46]. Almost all of Scotland's primary forests have been cleared, and peatlands are widespread on the moors and hills, which are largely used for sheep farming and deer and grouse estates [45, 46]. Only 10% of the UK's population live in Scotland, 75% of which dwell in the Central Lowlands, leaving rural areas sparsely populated [45].

### Rationale

Wolf habitat must be considered at a landscape scale, due to the size of the pack territories, which are typically 100 – 200km$^2$, but vary greatly [16, 20, 47], and due to wolves' long-distance dispersal, which can be hundreds of kilometres [47]. Many European studies find there is a strong correlation between wolf presence and forest cover [17, 18, 22, 23], but it must also be recognised that in these countries, areas with *low* human influence and *high* prey density tend to have correspondingly *high* forest cover. For instance, the Swiss Valais is 22% forest,

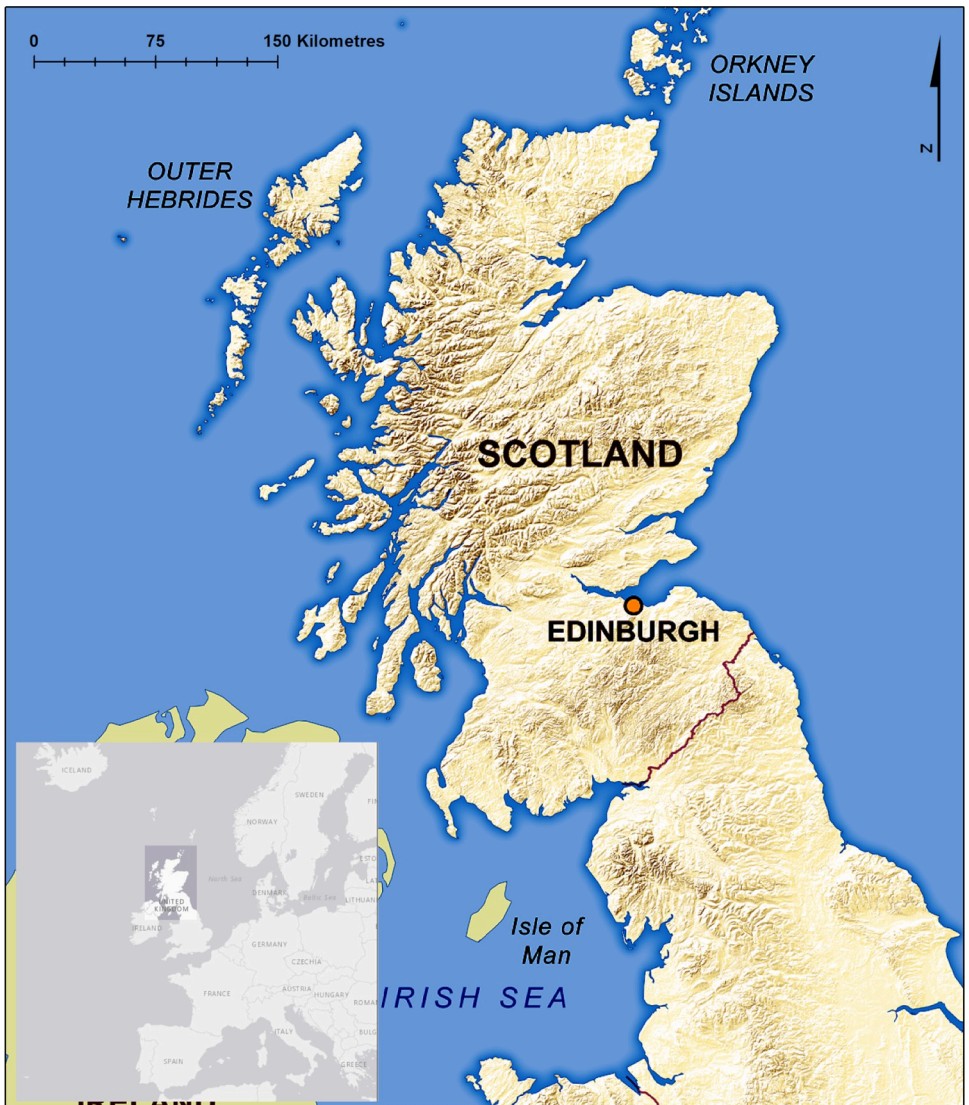

**Fig 1. Physical map of Scotland and its location within north-west Europe inset.** The mountains of the Highlands/
Grampians in the North, the belt of the Central Lowlands, and the hills of the Southern Uplands can be clearly seen
(Ordnance Survey, 2013, © Crown copyright and database rights 2022 Ordnance Survey (100025252); openstreetmap.
org, n.d., © OpenStreetMap contributors.).

and nearly all ungulates are restricted to forest habitats, especially in winter [21]. Jędrzejewski
et al. [22, 23] also attribute Polish wolf pack preference for forest cover to avoidance of
humans. This association between high forest cover and high prey density/low human pres-
ence is not the case in Scotland, which is only 18.5% forest (mostly conifer plantations) and
where heathland and upland bog constitute the majority of its unpeopled and deer-stocked
wild lands [48, 49]. This is important, because it means that non-forested habitats in Scotland
may well be of greater value to wolves than they might initially appear. Meanwhile, many
American models find that wolf presence and abundance is more directly related to prey den-
sity or human-caused mortality risk than land cover [16, 19, 24]. For instance, the wolf packs
in the Canadian Arctic follow the caribou herds regardless of habitat [50]. Similarly in north-
eastern USA, prey availability and not habitat type explained 72% of spatial wolf population

variation [19]. Road density is also recognised as a crucial factor in habitat suitability in multiple studies (e.g. [51, 52]). This difference between American and European studies suggests that either different limiting factors are at play, or that high covariance makes it hard to disentangle the importance of each variable, or that wolves are adaptable and therefore their habitat can be characterised by different variables in different places.

Similarly, slightly different wolf predation behaviours emerge from different European studies, though wild ungulates always predominate (though see Ciucci et al. [17], for scavenging behaviour on garbage dumps) [47, 53]. Though roe and red deer form the majority of wolf diets in most European studies, some studies suggest red deer are preferred (though roe deer often still make up the majority of the diet due to higher availability) [22, 54–58]. Therefore, it is likely that in Scotland, both roe and red deer would be predated, though there may be a preference for red deer where they are available. Fallow and Sika deer are also found in Scotland, but little data exists for wolf predation on these species [59].

Despite these variations across studies, land cover, prey density, road density and human density emerge as the most important factors in wolf habitat suitability. As regards land cover, we noted which cover types are associated with wolf presence and absence, but as regards the other three variables–which are continuous rather than categorical variables–we needed to establish thresholds of suitability and unsuitability.

Prey densities that characterise areas of wolf presence (i.e. suitable habitat) vary across studies. Jędrzejewski et al. [23] noted a drop-off in wolf presence only when prey densities were as low as 0.6 deer/km$^2$, but other studies find density requirements of at least 4/km$^2$, with up to 13 elk per km$^2$ recorded in the wolf ranges in Yellowstone National Park [13, 24, 25, 53, 60].

As regards roads, road density (km/km$^2$) is the standard metric used in studies that assess wolf responses to roads [16, 19, 21–23, 51, 52]. Recorded road densities in areas of wolf presence (i.e. suitable habitat) vary from 0.2km/km$^2$ to around 0.4 or 0.5km/km$^2$. Though there are areas with road densities of 0.7km/km$^2$ being resettled by wolves, most studies thus far find such densities to be largely unsuitable [16, 19, 21, 51, 52, 61].

Recorded human densities in areas of wolf presence (i.e. suitable habitat) begin at 0.43/km$^2$ [24], but there is some variation in where the upper limit lies, with the same study finding an average of just 2.33 people/km$^2$ in non-pack areas, whereas other studies record human densities all the way up to 36.7/km$^2$, especially in Europe [16, 21, 25, 61–63].

Given these varying preferences and behaviours in different regions (none of which are entirely comparable to Scotland), it may seem challenging to derive suitability rules that would apply to Scotland. However, the wolf's generalist ecology helps to offset this. Wolves are not habitat specific, and nor are they necessarily wilderness species. They have colonised habitats throughout the northern hemisphere wherever they are protected from persecution, from 20˚ north up to the Arctic [3, 16, 64]. Wolf core ranges have been found to include a wide range of habitats in addition to forests, including pasture, chaparral, eskers, heath tundra, and human garbage dumps [17, 50, 65]. Recently wolves successfully recolonised a National Park in the Netherlands, an urbanised country with an average human population density of 512/km$^2$ (against Scotland's 70/km$^2$) [66, 67]. Additionally, Scotland was until recently (ecologically speaking) part of the wolf's range, and it was eradicated by persecution rather than by a lack of suitable habitat. Of course it may be fallacious to assume that because Scotland offered suitable wolf habitat in the past, it continues to do so now and in the future, but this only increases the need for rigorous study to test if that is so [32]. Osborne and Seddon [32] recognise that it is essential to model extensively before reintroduction of any species, and that unsuitable habitat may be the main reason for reintroduction failures in the past.

Once the most important factors had been identified, and suitability thresholds established for each, fuzzy membership was applied to GIS datasets of the three continuous variables

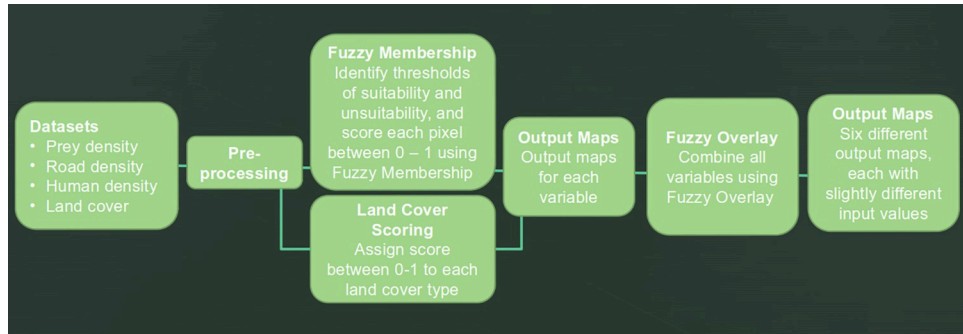

**Fig 2. Flowchart of process.** A summary of the analysis process, from input datasets to output maps. Note there are output maps for each variable individually, and then further output maps of all four variables combined using fuzzy overlay.

across the Scottish mainland, while land cover types were allocated scores and likewise mapped. The resulting output maps were combined using fuzzy overlay (Fig 2). This process was applied to six variations in input data, to explore uncertainty and to test sensitivity. The result is a set of six output maps, each containing all four variables, grading the Scottish mainland according to its suitability as wolf habitat.

## Datasets

Spatial datasets for the four variables for Scotland were assembled in a GIS (ArcGIS Desktop 10.7.1 [68]), clipped to the Scottish mainland and converted to raster with a resolution of 500m x 500m pixels if originally in vector format (Table 1).

**Land cover.** The Corine Land Cover map is a European land cover inventory in 44 classes, based on Sentinel and Landsat imagery. It has a resolution of 25ha, and is classified at three levels of increasing thematic detail (e.g. the land cover type "Wetlands" is subclassified into "Coastal Wetlands" and "Inland Wetlands", which are themselves subclassified into five further classes). The middle level of classification was used, as it had an appropriate level of thematic resolution. Each land cover type was scored for wolf suitability according to the available literature on wolf habitat preferences described in the Rationale (Table 2). A similar approach was used by Sandom et al. [13] in their modelling of a hypothetical fenced reserve in Scotland, but here an index between 0 and 1 was used, as that is comparable to the way fuzzy membership is allocated (where 0 indicates unsuitable habitat, and 1 suitable habitat).

**Table 1. Datasets and their sources used in the analysis of habitat suitability.**

| Variable | Source | Date of data collection/ dataset creation |
|---|---|---|
| Land cover | Corine Land Cover vector map, 2020 version https://land.copernicus.eu/pan-european/corine-land-cover/clc2018?tab=mapview | 2018 |
| Prey density | Scottish Natural Heritage Deer Count Density vector map, revised 2018 https://gateway.snh.gov.uk/natural-spaces/dataset.jsp?dsid=DCD | 2010 |
| Road density | Ordnance Survey's Open Roads vector map https://www.ordnancesurvey.co.uk/business-government/products/open-map-roads | 2020 |
| Human density | National Records of Scotland (2011). 2011 Census: boundary data vector map https://beta.ukdataservice.ac.uk/datacatalogue/studies/study?id=5819&type=Data%20catalogue | 2011 |

**Table 2. Land cover suitability scores.**

| Land cover | Score |
|---|---|
| Arable land | 0 |
| Artificial, non-agricultural areas | 0 |
| Coastal wetlands | 0.4 |
| Forest | 1 |
| Heterogeneous agricultural areas | 0.2 |
| Industrial, commercial and transport units | 0 |
| Inland waters | 0 |
| Inland wetlands | 0.2, 0.4 & 0.6[a] |
| Marine waters | 0 |
| Mine, dump and construction sites | 0 |
| Open spaces with little or no vegetation | 0.2, 0.4 & 0.6[a] |
| Pastures | 0 |
| Permanent crops | 0 |
| Shrub and/or herbaceous vegetation associations | 0.2, 0.4 & 0.6[a] |
| Urban fabric | 0 |

[a]These land cover types were scored three times due to uncertainty about their level of suitability.

The suitability of the land cover types "Inland wetlands", "Open spaces with little or no vegetation", and "Shrub and/or herbaceous vegetation associations" was particularly hard to score, because these land cover types are uncommon in areas where other wolf habitat suitability studies have been performed, and thus their suitability is unclear. Additionally, a large proportion of Scotland's red deer population roams these open habitats, which is in contrast to research in Europe, where ungulates are largely confined to forests [49]. Their level of suitability is also crucial as they are dominant habitats in Scotland. Therefore, the model was run once with them scored at 0.2, once at 0.4, and once at 0.6, so that the model sensitivity to this particular variable could be explored. These scores can be considered to indicate substantially unsuitable, somewhat unsuitable and somewhat suitable habitat, respectively. Though Pastures may technically be suitable for wolves, they were valued at 0, because the inclusion of livestock pasture in proposed wolf territory could promote livestock predation and human-wildlife conflict.

**Prey density.** The map of deer density had a resolution of $1km^2$, with count data attached to each $1km^2$ cell. Due to the herding behaviour of red deer in the Highlands, deer density was highly aggregated, i.e. one cell could contain dozens of deer while those around it contained none, reflecting where the herd happened to be on the day of the count. As this snapshot did not accurately reflect the realised spatial density of deer over time, kernel density estimation (KDE) was applied to each herd location, with an output cell size of 50m and a search radius of 1480m (representing the average home range of a red deer in Scotland, where they mainly roam open heaths and peatlands) [49, 69]. This "smooths" the density over a wider area, in recognition of the fact that the herd will move around its range, and so the entire range may be considered to offer prey [70].

Data from multiple studies suggest roe deer are an important component in wolf diets (see Rationale). Unfortunately, almost no roe deer are included in SNH's Deer Count Density map, and no alternative roe deer density data was found. However, Campbell is cited as stating that roe deer density in the Highlands is $7.4/km^2$ and in the Southern Uplands $5.5/km^2$ [59]. Therefore analysis was performed once with only the SNH dataset, and then a second time in which

roe deer were additionally incorporated. They were incorporated at the densities mentioned above in every km$^2$ cell for the Highlands and Southern Uplands (in the council areas of Highland, Dumfries and Galloway, and Scottish Borders, to be precise). Higher deer density suitability thresholds were used in the second version to account for the smaller body size of roe deer. The values used were, as with the red deer, guided by existing literature on roe deer densities in wolf territories [25]. Due to the paucity of roe deer density data, this should be considered only as indicative of whether roe deer presence/absence might strongly change the outcome of the model, and more research is no doubt needed.

**Road density.** Road density (km/km$^2$) was calculated from the Open Roads vector map at a resolution of 0.5km$^2$. This dataset includes minor roads but not private roads, and no weighting was applied to roads of different rank, as this does not seem to be common practice in wolf/road studies. However, this could be worthy of further investigation.

**Human density.** Human density (people/km$^2$) was calculated from the boundary census data vector map and converted to raster. With population data available only at census boundary scale, this is the dataset with the coarsest resolution. These boundaries are small in urban areas, where the population is high, but large in rural areas.

## Processing

Thresholds that represented completely suitable conditions (scored as 1) and completely unsuitable conditions (scored as 0) were established from the literature as regards deer, road and human densities (Table 3). Fuzzy membership with a linear membership type was then applied to each of these datasets accordingly, resulting in maps showing the suitability of that variable for wolves across mainland Scotland. This process was repeated for the deer density dataset with additional roe deer densities incorporated. The land cover dataset could not be processed using this method, because although the suitability scoring applied is numeric, it is categorical rather than continuous data, and therefore has no marginal cases [30].

All four outputs were then assembled using fuzzy overlay with overlay type gamma of 0.9. Due to the paucity of fuzzy analysis in habitat suitability studies, there was no justification for using a different value, but this could be investigated further. Because of the two versions of deer density and three versions of land cover datasets in use, this resulted in six output maps. These maps each incorporate all four variables, but with some changes in input values in the deer density and land cover variables (Table 4).

The fuzzy overlay maps were reclassified into 10 equal classes of suitability using ArcMap's Reclassify tool. The distribution of pixels across the classes could then be used to calculate area and proportion of the Scottish mainland falling into each class, i.e. what proportion of land is 0–10% suitable, 10–20% suitable, and so on. Finally the Create Random Points tool and the Sample tool (resampling technique: nearest) were used to extract cell values from the same 500 randomly-generated points on each of the 6 fuzzy overlays, and a test for difference was performed in SPSS. As the data was strongly non-normal (Shapiro-Wilk: p = <0.001), Kruskal-Wallis with all pairwise comparisons was used.

**Table 3. Thresholds of suitability.**

| Variable | Suitable | Unsuitable |
|---|---|---|
| Prey density without roe deer | $> = 5.5$/km$^2$ | $< = 1$/km$^2$ |
| Prey density with roe deer | $> = 7$/km$^2$ | $< = 3$/km$^2$ |
| Road density | $< = 0.23$km/km$^2$ | $>0.7$km/km$^2$ |
| Human density | $< = 2$/km$^2$ | $> = 37$/km$^2$ |

**Table 4. Versions of land cover dataset and prey density dataset used in each model.**

| | Inland wetlands, Open spaces with little or no vegetation, and Shrub and/or herbaceous vegetation | | |
| --- | --- | --- | --- |
| | Suitability score = 0.2 | Suitability score = 0.4 | Suitability score = 0.6 |
| **Without roe deer** | Model 1 | Model 2 | Model 3 |
| | (Fig 3A) | | (Fig 3B) |
| **With roe deer** | Model 4 | Model 5 | Model 6 |
| | (Fig 3C) | | (Fig 3D) |

## Results

The maps of land cover fuzzy membership analysis (Fig 3A–3C) show forest consistently as bright green, i.e. suitable. The large areas that vary from brown to orange to yellow, depending on the model used, correspond to the three land cover types on which there is little suitability data: Inland wetlands, Open spaces with little or no vegetation, and Shrub and/or herbaceous vegetation associations. These maps make it explicit what a large area is covered by these three habitats, and therefore how important their level of suitability is.

Meanwhile, it can be seen that deer densities are largely suitable in the Highlands and Grampians (Fig 3D and 3E). The addition of a roe deer baseline density in three council areas increases the suitability of the Highlands and Southern Uplands. Suitable road densities (Fig 3F) are similarly limited to the Highlands and Grampians, but human density (Fig 3G) is suitable for wolves across a large proportion of Scotland.

Fig 4 shows overall habitat suitability for wolves when all four variables are combined, as per four of the six models (Table 4). The more suitable habitat is concentrated in the Highlands and Grampian Mountains in all model outputs. The two fuzzy overlay maps using the three land cover types scored at 0.4 can be seen in the Supporting Information (S1 Fig).

Calculating the area and percentage of mainland Scotland that falls into ten equal classes of suitability (Table 5) shows that the majority of Scotland is unsuitable according to these models, and this does not vary much between models. There is more variation at the high suitability end of the scale, with between 0.6% and 21% of the area (or between 384km$^2$ and 14259.5km$^2$) rated most suitable (0.9–1.0), depending on the model used. Regardless of the model used, land is concentrated at either end of the scale of suitability, with very little semi-suitable habitat.

Plotting this in graph form makes it plain that the different models have little impact on the distribution of land within the less suitable categories, but a larger impact on distribution in the more suitable categories (Fig 5). It also becomes evident that the changes in land cover scoring have more of an impact on results than the addition of roe deer baseline densities.

Testing for difference in a sample of 500 randomly-generated cell values taken at the same points for all six fuzzy overlay maps shows that there is only a significant difference in results between Model 1 and Model 6, i.e. the most and least "generous" models (H = -3.47, df = 5, p = 0.008, using the adjusted significance value).

## Discussion

Our results have shown that there is a high level of covariance between three of the variables, with the most suitable areas in terms of prey density, road density and human density all concentrated in the same regions. This results in the Highlands and Grampian mountains emerging strongly and consistently as the areas most suitable for wolves in mainland Scotland. Though this area is contiguous, it is bisected by the A82 and the many lakes of the Great Glen, which could be barriers to wolf movement. Human density, prey density and road density are

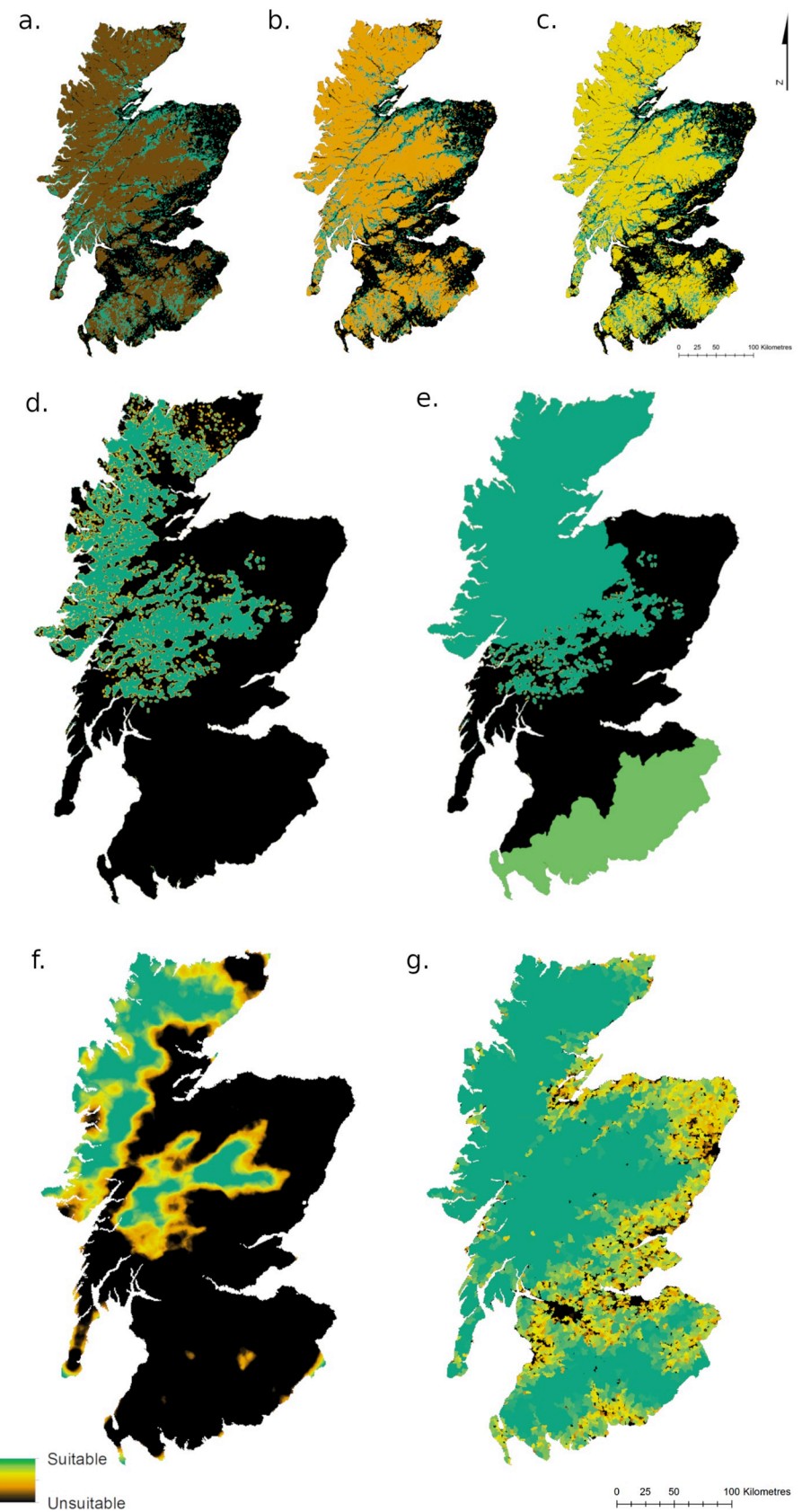

**Fig 3. Fuzzy membership output maps.** These show the suitability of the four variables: land cover (a,b,c) with three key types scored at 0.2, 0.4 and 0.6 respectively; prey density (d,e) without and with roe deer incorporated as a baseline density in two areas; road density (f) and human density (g). The Highlands and Grampians emerge consistently as the most suitable areas in all variables.

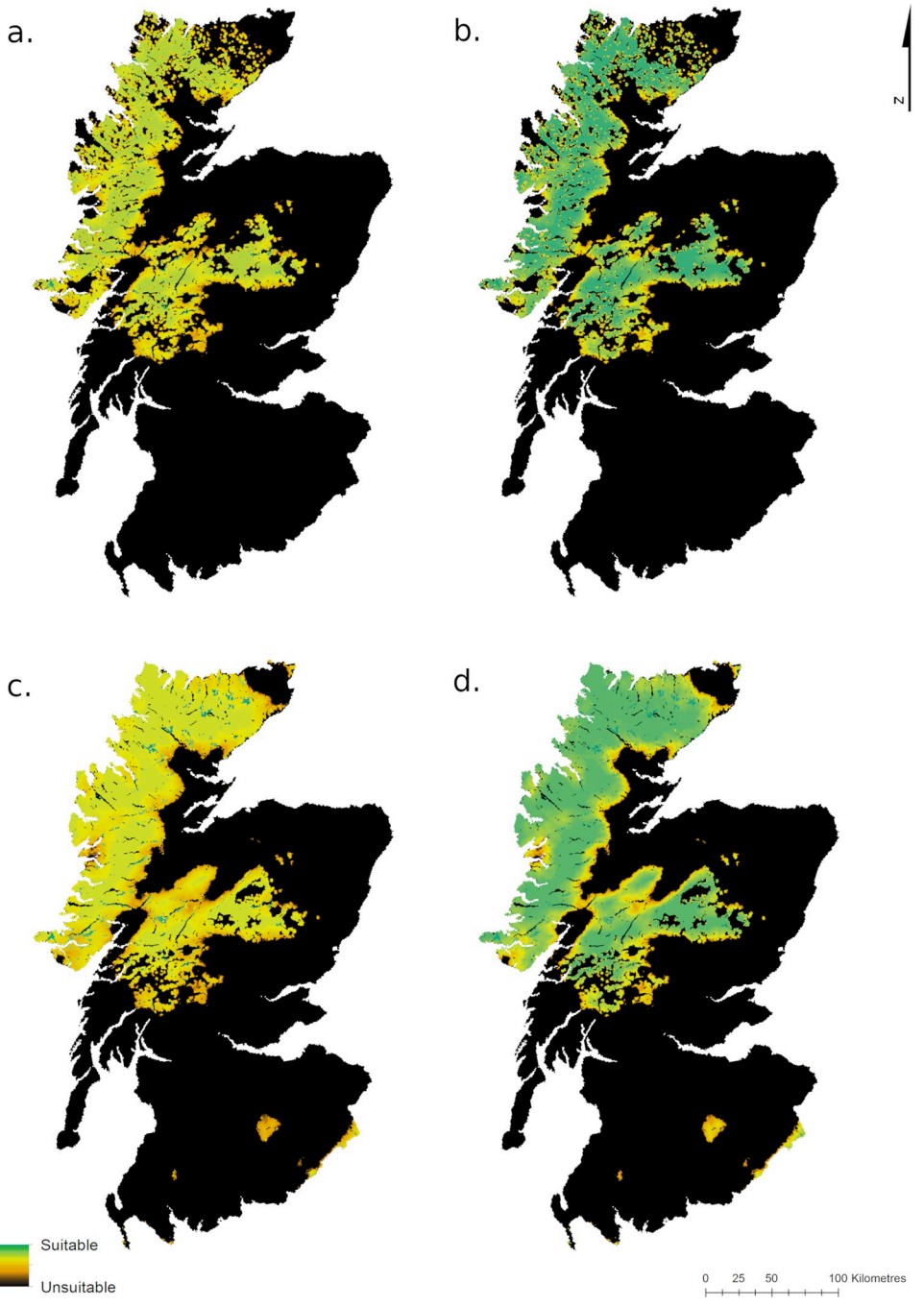

**Fig 4. Overall wolf habitat suitability.** Fuzzy overlay maps combining all four variables. Fig 4A corresponds to Model 1, Fig 4B to Model 3, Fig 4C to Model 4 and Fig 4D to Model 6 (Table 4).

**Table 5. Area of Scotland by suitability class.**

| Suitability | | Model 1 | Model 2 | Model 3 | Model 4 | Model 5 | Model 6 |
|---|---|---|---|---|---|---|---|
| 0.0–0.1 | Area (km$^2$) | 51980.5 | 51981.5 | 51980.0 | 47305.5 | 47305.5 | 47306.0 |
| | Percentage | 76.5 | 76.5 | 76.5 | 69.7 | 69.7 | 69.7 |
| 0.1–0.2 | Area (km$^2$) | 0.0 | 0.0 | 0.0 | 0.0 | 0.0 | 0.0 |
| | Percentage | 0.0 | 0.0 | 0.0 | 0.0 | 0.0 | 0.0 |
| 0.2–0.3 | Area (km$^2$) | 1.0 | 0.5 | 0.5 | 0.0 | 0.0 | 0.0 |
| | Percentage | 0.0 | 0.0 | 0.0 | 0.0 | 0.0 | 0.0 |
| 0.3–0.4 | Area (km$^2$) | 15.5 | 12.0 | 6.5 | 9.5 | 5.5 | 4.0 |
| | Percentage | 0.0 | 0.0 | 0.0 | 0.0 | 0.0 | 0.0 |
| 0.4–0.5 | Area (km$^2$) | 86.5 | 46.0 | 37.0 | 58.0 | 30.0 | 20.0 |
| | Percentage | 0.1 | 0.1 | 0.1 | 0.1 | 0.0 | 0.0 |
| 0.5–0.6 | Area (km$^2$) | 346.0 | 200.0 | 146.0 | 225.0 | 139.5 | 101.0 |
| | Percentage | 0.5 | 0.3 | 0.2 | 0.3 | 0.2 | 0.1 |
| 0.6–0.7 | Area (km$^2$) | 1244.0 | 691.5 | 492.0 | 953.0 | 503.5 | 351.5 |
| | Percentage | 1.8 | 1.0 | 0.7 | 1.4 | 0.7 | 0.5 |
| 0.7–0.8 | Area (km$^2$) | 4097.5 | 2114.0 | 1492.0 | 4131.5 | 1917.5 | 1270.5 |
| | Percentage | 6.0 | 3.1 | 2.2 | 6.1 | 2.8 | 1.9 |
| 0.8–0.9 | Area (km$^2$) | 9755.5 | 7076.0 | 4237.0 | 14477.5 | 8820.5 | 4597.0 |
| | Percentage | 14.4 | 10.4 | 6.2 | 21.3 | 13.0 | 6.8 |
| 0.9–1.0 | Area (km$^2$) | 384.0 | 5791.5 | 9519.0 | 750.0 | 9190.0 | 14259.5 |
| | Percentage | 0.6 | 8.5 | 14.0 | 1.1 | 13.5 | 21.0 |

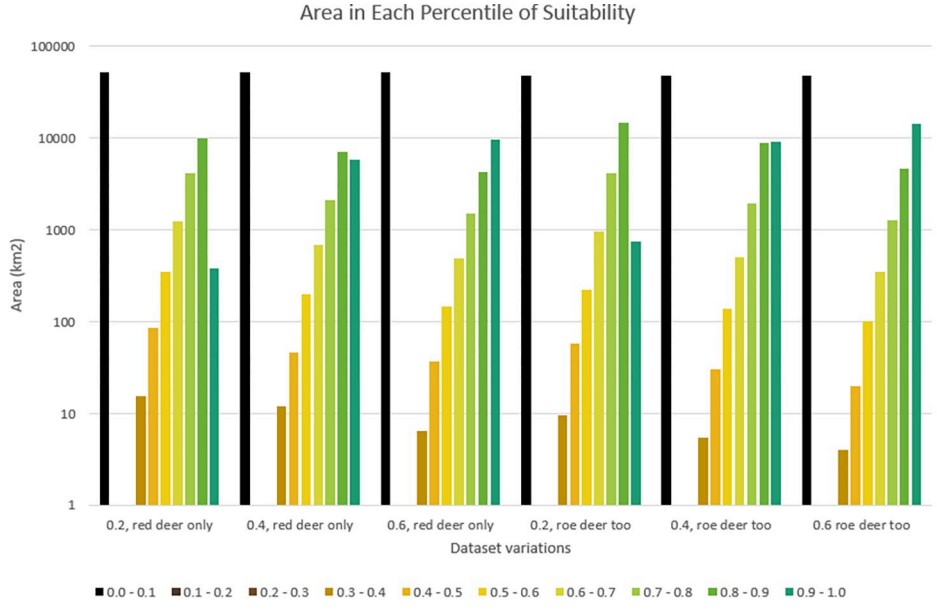

**Fig 5. Area of Scotland by suitability class.** Area of the Scottish mainland that falls into each of ten classes of suitability under different models (models 1–6 from left to right). Note that the Y-axis uses a logarithmic scale. All models show a similar pattern, with the distribution being highly uneven, and most areas being either completely unsuitable or substantially/completely suitable.

suitable throughout this region, and the addition of roe deer to the prey density map makes little difference. This is partly because in the Highlands the high densities of red deer already reach the suitability threshold, whereas in the Southern Uplands, the majority of the region is excluded anyway due to high road densities. It is also partly because the adjustment of the suitability thresholds upwards to account for the smaller body size of roe deer somewhat negates the gains of including them.

However, the suitability of the fourth variable, i.e. land cover, depends heavily on how suitable open heath and bog habitats are for wolves. It is the scoring of three land cover types in this variable that make the largest difference in the fuzzy overlay maps. Though in all cases the Highlands and Grampians still emerge as most suitable, differences in scoring mean they may be anywhere between somewhat suitable and completely suitable (between 0.7 and 1.0).

In terms of sensitivity, it can thus be concluded that the model is not particularly sensitive to the changes in prey density used here. However, it is somewhat sensitive to changes in land cover scoring: though the regions with highest suitability do not change, their level of suitability does.

As regards prey density, road density, and human density, the models could be considered relatively conservative. This is due to two reasons: the estimates of suitability and unsuitability adopted as thresholds were conservative; and prey density is likely to be higher and more widespread than the SNH's deer counts suggest, as these counts mostly only include red deer spotted in open areas where and when a count is carried out. The British Deer Society's distribution survey [71] finds that red deer are extant also in the Southern Uplands (which is not represented in the SNH counts), roe deer are common across Scotland, and fallow and sika deer are also found patchily in the Cairngorms, Highlands, west Scotland and the central Southern Uplands. However, it must be noted that regardless of prey density much of Scotland would remain unsuitable for wolves due to high road densities (Fig 3F).

Deer densities far exceed the threshold of suitability in much of the Highlands and Grampians with many areas holding >35/km$^2$ according to SNH's deer counts (after KDE processing). Hetherington and Gorman [59] quote an average density of 12.2 deer of all species/km$^2$ in the Highlands, while 11–12/km$^2$ is estimated by Sandom et al. [13]. This is significant because wolf pack range size is largely determined by prey availability [19, 47, 53]. Recorded range sizes vary from 33km$^2$ to 6272km$^2$ but average 100 – 200km$^2$ [16, 20, 47, 61]. Fuller [53] found that at a density of 6.2 white-tailed deer per km$^2$ (roughly half the Highland deer density) pack range size was only 116km$^2$. Meanwhile, Sandom et al.'s [13] modelling suggested that a Highland fenced reserve of 600km$^2$ would sustain 2 packs of 4 wolves for at least 100 years. At their most conservative, the models used in this study place an area of 10,139km$^2$ between 80% and 100% suitable, and at their most generous, 18,857km$^2$ is 80–100% suitable. This suggests that there may be sufficient wolf-suitable area to support between 50 and 94 packs of 4 wolves, if pack territory size is taken to be 200km$^2$. However, it should be noted that the minimum size required for a single pack is unclear due to variations in range size recorded in the literature, and Sandom et al. [13] found that a fenced reserve of 200km$^2$ was too small to support a Highland wolf pack for 100 years (though fencing brings with it implications that an unfenced population would not face). Additionally, a single–or even several–packs is not a self-sustaining population, as evidenced by the isolated wolf population on 544km$^2$ Isle Royale in Lake Superior, whose numbers dwindled from 50 in 1980 to 2 in 2016 before reintroductions bolstered them [72]. Indeed in 1992, the US Fish and Wildlife Service stated that an area of 25,000km$^2$ was required for a self-sustaining wolf population [19].

However, the evidence base of what wolves require is still developing. Linnell et al. [64], Mech [3], and Mladenoff et al. [16] all note that with protection from persecution, wolves are recolonising areas previously thought unsuitable due to high human and road densities. Future

observations of recolonising wolves in Europe are likely to be instructive as to what wolves prefer and tolerate. Clarification on wolf preferences as regards open upland habitats is particularly needed for assessment of Scottish habitat suitability, as studies currently conflict on how essential forests are for wolves. It is likely that such open habitats are suitable for wolves in Scotland, due to their high prey density, but they may offer less shelter and fewer denning opportunities. Though this is the largest data gap as regards modelling Scottish habitat suitability, other areas of future research could also include wolf response to roads. The standard in wolf habitat studies is employing a road density measure of km/km$^2$. However, Jędrzejewski et al. [23] found that Polish wolves avoided a 250m wide belt along roads, so the use of buffer zones in models may be more beneficial. Additionally, there seems to be little research on the implications of roads of different class, and this could be explored further. Further research into Scottish habitat suitability would also benefit from more comprehensive deer density datasets, or else modelling of deer populations across Scotland that is more sophisticated than the KDE smoothing used here. Reinecke et al. [70] point out that one of the weaknesses of KDE is that it includes invalid areas (for instance a loch), and suggest minimum convex polygons or α-local convex hulls as alternative methods for modelling red deer. Dasymetric interpolation may also provide a more realistic model of deer densities [73]. However, as our sensitivity testing indicated that changes in deer density did not particularly affect the model output, we did not refine our processing for this study.

There are many considerations regarding the return of wolves to Scotland that are beyond the scope of this study. These include the requirements of maintaining wolf genetic diversity and metapopulation, which would require either an area far bigger than that needed to support a few packs, or else regular introductions of additional animals, though this may result in conflict with existing packs. There are also implications arising from wolf dispersal (which can be many hundreds of kilometres) and social ecology (which is complex and could be negatively affected in a small, constrained population) [47]. These implications are not explored here, but would be worth further study. While this study finds deer densities are easily sufficient to support wolves in the Highlands and Grampians, it does not model the long-term predator-prey relationship or the likely impact on deer population dynamics. Lastly, and as discussed in the Introduction, one of the most important considerations in any reintroduction project is public attitudes and impacts on local communities. This is especially true in terms of livestock predation, but wolf reintroduction also has the potential to affect the recreational value of Scotland's wild lands, and even its Protected Areas. This essential factor would require extensive further research and public consultation, and high levels of support would need to be demonstrated before any reintroduction could be considered. The recent consultation on lynx reintroduction by Lynx to Scotland may provide a good practice model [74].

For further research on wolves in Scotland see Sandom et al.'s [13] ecological feasibility study into a fenced reserve for wolves in Scotland, in which they conclude that there is suitable area in the Highlands for a 600km$^2$ reserve, that this would be sufficient to support a functional wolf population, and that such a population has the potential to regulate deer numbers. See Nilsen et al.'s [1] modelling of predator-prey dynamics and ecological impacts which suggests wolf reintroduction would have both financial and ecological benefits in Scotland, due to the reduction in deer numbers. They also explore public attitudes and perceptions and find these to be largely positive, though negative amongst farmers. Finally see Wilson's [10] review of research on large carnivores, which also finds that the Highlands could support a viable wolf population. He also reviews risks to humans and livestock, and finds healthy wolves appear to pose no risk to humans, but that livestock predation does occur, though wild prey is preferred where available. Lastly, he finds that attitudes to reintroduction are usually positive amongst

the general public, but negative amongst those most likely to suffer losses, and concludes that public support for reintroduction in the UK is probably currently lacking.

With the recolonisation of Europe by wolves, Britain increasingly becomes an outlier in its lack of apex predators. If the nation were not an island, wolves would likely soon cross our borders, if they had not already done so. As it is, short of escape from captivity, it is impossible for wolves to recolonise naturally, regardless of the suitability of our habitats or the desirability of their presence. Whether they return to Britain is a decision we must make actively, and in full consideration of the wolf's requirements and impacts. Therefore conservationists need to anticipate evidence needs now [2]. As well as the need to fill knowledge gaps (some of which are identified above) both Manning et al. [2] and Sandom et al. [13] advocate reintroduction experiments in large-scale Highland enclosures, which would allow us to discover the impacts of wolves on deer and ecosystems in general, in the context of the Highlands. This study supports previous conclusions that in terms of habitat suitability, the Highlands or Grampians would be the most appropriate places for wolf reintroduction. We recommend further research into the knowledge gaps outlined, and beyond that–should Scotland still appear suitable for wolves, as we find–consideration be made of the desirability of reintroduction.

## Conclusion

This study set out to identify the level of habitat suitability for wolves in the Scottish mainland. We have established thresholds of suitability and unsuitability from the literature as regards the four most important habitat variables: land cover, prey density, road density and human density. We mapped each variable according to its suitability across mainland Scotland using fuzzy membership analysis, and then combined all the variables into maps of overall suitability using fuzzy overlay. The Highlands and the Grampians emerged strongly as the most suitable areas, and sensitivity testing showed the model was fairly robust to changes in the prey density inputs, but less robust to changes in the land cover inputs, which we identify as an area in need of further research. Between $10,139km^2$ and $18,857km^2$ are found to be 80–100% suitable, depending on the model used, which may be sufficient to support between 50 and 94 packs of 4 wolves.

## Supporting information

**S1 Fig. Habitat suitability maps produced using the three land cover types scored at 0.4.** These were produced by Model 2 (a) and Model 5 (b) in Table 4, respectively. (TIFF)

## Author Contributions

**Conceptualization:** Vashti Gwynn, Elias Symeonakis.

**Data curation:** Vashti Gwynn.

**Formal analysis:** Vashti Gwynn.

**Methodology:** Vashti Gwynn, Elias Symeonakis.

**Project administration:** Vashti Gwynn, Elias Symeonakis.

**Resources:** Vashti Gwynn.

**Supervision:** Elias Symeonakis.

**Validation:** Vashti Gwynn.

**Writing – original draft:** Vashti Gwynn.

**Writing – review & editing:** Vashti Gwynn, Elias Symeonakis.

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
