## [Decision Letter · Decision Letter 0]

30 Jun 2022

PONE-D-22-05836Rule-based habitat suitability modelling for the reintroduction of the grey wolf (*Canis lupus*) in ScotlandPLOS ONE

Dear Dr. Gwynn

Thank you for submitting your manuscript to PLOS ONE. After careful consideration, we feel that it has merit but does not fully meet PLOS ONE’s publication criteria as it currently stands. Therefore, we invite you to submit a revised version of the manuscript that addresses the points raised during the review process.

I agree with the two referees that appreciate your work and I think that with some amendments it can be suitable for publication. Please follow carefully the suggestions you have received with special care to define a  profile and relative acceptance of the idea of wolf reintroduction by rural and urban part of human population, including the information about public knowledge,(Table 2). Moreover, please consider the idea suggested by Alberto Meriggi about a better evaluation of suitability scores of the classes.

We look forward to receiving your revised manuscript.

Kind regards,

Marco Apollonio

Academic Editor

PLOS ONE

Journal Requirements:

2. We note that Figure 1 and 2 in your submission contain copyrighted images. All PLOS content is published under the Creative Commons Attribution License (CC BY 4.0), which means that the manuscript, images, and Supporting Information files will be freely available online, and any third party is permitted to access, download, copy, distribute, and use these materials in any way, even commercially, with proper attribution. For more information, see our copyright guidelines: http://journals.plos.org/plosone/s/licenses-and-copyright.

1. You may seek permission from the original copyright holder of Figure 1 and 2 to publish the content specifically under the CC BY 4.0 license. 

3. We note that you have stated that you will provide repository information for your data at acceptance. Should your manuscript be accepted for publication, we will hold it until you provide the relevant accession numbers or DOIs necessary to access your data. If you wish to make changes to your Data Availability statement, please describe these changes in your cover letter and we will update your Data Availability statement to reflect the information you provide

Reviewers' comments:

Reviewer's Responses to Questions

**Comments to the Author**

1. Is the manuscript technically sound, and do the data support the conclusions?

Reviewer #1: Partly

Reviewer #2: Yes

2. Has the statistical analysis been performed appropriately and rigorously? 

Reviewer #1: Yes

Reviewer #2: Yes

3. Have the authors made all data underlying the findings in their manuscript fully available?

Reviewer #1: Yes

Reviewer #2: Yes

4. Is the manuscript presented in an intelligible fashion and written in standard English?

Reviewer #1: Yes

Reviewer #2: Yes

5. Review Comments to the Author

Reviewer #1: Authors have taken novel approach to frequent conservation topic of habitat suitability and have done a good job considering the (un)availability of data. However, one part of data could be included, if not in the modelling, but at least as part of the description of landscape and in the discussion. It is the information about livestock husbandry, density of livestock and policy. The second aspect is human attitude toward wolves and it could be connected with the livestock topic. It could have been an nice addition to the study, as another habitat variable, maybe providing a profile and relative acceptance of the idea of wolf reintroduction by rural and urban part of human population, including the information about public knowledge, believes attitudes and opinions regarding wolf reintroduction. This could improve the modelling. Biologist rarely like to do social science, but when it is about wolves, this aspect is often more relevant than ecology.

My other comments are in the attached PDF

Kind regards,

J. Kusak

Reviewer #2: Revision of the manuscript PONE-D-22-05836

The manuscript by V. Gwynn and A. Symeonakis "Rule-based habitat suitability modelling for the reintroduction of the grey wolf (Canis lupus) in Scotland" proposes a new method for modelling the habitat suitability of Scotland for the wolf, based on fuzzy logic. The proposed method appears convincing because in the case of Scotland, given the time elapsed since the extinction of the species, it is not possible to start from certain presence data to estimate the habitat suitability. Furthermore, the variables that can contribute to determining the environmental suitability are also measurable with a certain degree of uncertainty. The results obtained are also convincing and the analyses carried out are based on an in-depth analysis of the available literature. The manuscript appears clear and straigtforward in all its parts. The introduction is sufficiently thorough, the study area and the methods are described in detail, and the discussion is complete and relevant to the results. In the results, it would be better not to repeat the content of the captions of the figures but rather to illustrate what they show.

From a methodological point of view, I have only a few doubts concerning in particular the suitability scores attributed to the land use classes, especially pastures and open areas without vegetation (Table 2). Pastures can be largely used by wolves in particular when they host several livestock species especially sheep and goats. On the other hand, open spaces without vegetation shouldn’t be suitable for wolves. In fact, being without vegetation they should not be habitats rich in potential prey and, moreover, they do not offer shelter or the possibility of denning. The Authors should justify these choices.

Other comments directly on the manuscript

Alberto Meriggi

University of Pavia

6. PLOS authors have the option to publish the peer review history of their article (what does this mean?). If published, this will include your full peer review and any attached files.

Reviewer #1: **Yes: **Josip Kusak

Reviewer #2: **Yes: **Alberto Meriggi

---

## [Author Response · Author response to Decision Letter 0]

9 Sep 2022

Editor,

PLOS ONE

Carlyle House

Carlyle Road

Cambridge, CB4 3DN

United Kingdom

Manchester, 5th September 2022

Dear Editor,

Please find accompanying our revised manuscript entitled Rule-based habitat suitability modelling for the reintroduction of the grey wolf (Canis lupus) in Scotland. We would like to thank the Academic Editor, Professor Marco Apollonio, and the reviewers, Professor Josip Kusak and Professor Alberto Meriggi, for their careful reading and constructive comments. We provide below an explanation of how we have addressed each of the comments received. We hope we have successfully addressed all the reviewers’ concerns, and that the paper is now ready for publication. 

Please note that line numbers in our responses to the reviewers’ comments refer to the file with tracked changes, and will be different in the file with all changes accepted.

Yours faithfully,

Vashti Gwynn

On behalf of the authors

From Editor:

1) Check manuscript meets PLOS ONE’s style requirements, including those for file naming.

Checked and amended where necessary.

2) Figure 1 & 2 contain copyrighted images.

Figure 1 contains both Ordnance Survey and Open Street Map imagery. Use of both is covered by their licences (links below). Attributions in the Figure caption have been corrected, as per OS and OSM attribution policy. Figure 2 does not contain copyrighted images – I made this image myself.

Ordnance Survey licence - https://digimap.edina.ac.uk/help/copyright-and-licensing/os_eula/

Open Street Map licence - https://www.openstreetmap.org/copyright/en

3) We note that you have stated that you will provide repository information for your data at acceptance. Should your manuscript be accepted for publication, we will hold it until you provide the relevant accession numbers or DOIs necessary to access your data. If you wish to make changes to your Data Availability statement, please describe these changes in your cover letter and we will update your Data Availability statement to reflect the information you provide.

The geographic datasets we used are cited and are freely available. The dataset we created by randomly sampling the suitability values across our suitability model for the purposes of statistical analysis has already been uploaded to Zenodo repository (URL:

https://doi.org/10.5281/zenodo.6299108, DOI:10.5281/zenodo.6299108).

4) Upload figure files to the Preflight Analysis and Conversion Engine (PACE) digital diagnostic tool, https://pacev2.apexcovantage.com/ to ensure that figures meet PLOS requirements.

This has been done, and the converted figures have been uploaded with the corrected manuscript.

From Reviewer 1

General comments

1) One part of data could be included, if not in the modelling, but at least as part of the description of landscape and in the discussion. It is the information about livestock husbandry, density of livestock and policy.

We thank the Reviewer for their comment. This does indeed provide useful context, and its importance as a factor in reintroduction decisions should be emphasised. We have added an overview of Scottish farming and the implications for predator/wildlife conflict to our Introduction (line 141). We have also further emphasised in our Discussion the importance of this factor in any reintroduction decision (line 566).

2) The second aspect is human attitude toward wolves and it could be connected with the livestock topic. It could have been an nice addition to the study, as another habitat variable, maybe providing a profile and relative acceptance of the idea of wolf reintroduction by rural and urban part of human population, including the information about public knowledge, believes attitudes and opinions regarding wolf reintroduction. This could improve the modelling. Biologist rarely like to do social science, but when it is about wolves, this aspect is often more relevant than ecology.

We agree with the Reviewer’s comment. Likewise, this is important contextual information, and we recognise it is one of the most important considerations in any reintroduction. We have added a section profiling public attitudes to the Introduction (line 171), and we also now refer back to this topic and its importance in the Discussion (line 569).

For both 1) and 2), we have added to the Introduction an explanation as to why these factors are not included in our model (line 124), but to repeat here:

 • This is intended to be a habitat suitability study, i.e. an enquiry into where, if anywhere, in Scotland meets the wolf’s needs in terms of the physical characteristics of the landscape.

 • Attitudes to wolves, and our economic, social and psychological relationship with them are complex and varied. The subject deserves a more extensive treatment than we could give it in a habitat suitability study.

 • Attitudes, and economic, social and psychological impacts are dynamic, and opinions on rewilding and reintroduction are volatile (see Public Attitudes section). If these factors were incorporated into a habitat suitability model, that model would very quickly become outdated and therefore irrelevant.

The exclusion of farming and public attitudes from our model is not because we do not consider them important, but because we do not feel they are within the scope of a habitat suitability study. We hope the changes we’ve made better justify this choice, and represent the crucial important of these factors sufficiently.

Introduction

3) Where did wolves reach their “lowest point”? Europe? World?

Sentence amended to read “This persecution was part of a global eradication effort that brought overall wolf numbers to their lowest point between the 1930s and 1960s” (line 51).

4) References are needed for two statements: rewilding/reintroductions being currently popular, and beaver reintroduction receiving public attention and support.

Two citations have been added (line 83).

5) References needed for statement that Scotland has “extensive deer-filled wild lands and low human density”.

Citation added (line 91).

6) Why would land cover be an uncertain variable?

Sentence amended to read “We carry out a sensitivity analysis of the model for the variable whose suitability is most uncertain (land cover, due to a lack of data on the suitability of open habitats)” (line 117).

Methods

7) Suggestion to change “nation” to “country”.

This has been done (line 230).

8) Suggestion that differences between European and American habitat studies could also be due to wolves being adaptable.

Sentence amended to read “This difference between American and European studies suggests that either different limiting factors are at play, or that high covariance makes it hard to disentangle the importance of each variable, or that wolves are adaptable and therefore their habitat can be characterised by different variables in different places” (line 260).

9) Point that human attitudes were the reason for the original eradication of wolves, and that these would prove the most important factor in wolf survivorship if reintroduced.

Please see our response to point 2, above. We hope the additions to the text outlined there address this comment also.

10) Reiteration of the importance of human attitudes in light of the fact they were the previous cause of extinction.

Please see our response to point 2, above. We believe that the additions to the text outlined there address this comment, too.

11) What was the resolution of the converted rasters?

The resolution was 500m x 500m pixels. This has been added to the text (line 318).

12) Why not use DEMs or some derivations, such as terrain ruggedness, as variables?

Our research did not suggest that altitude or terrain, or related variables, were primary factors in wolf habitat suitability. In addition, the relationship between wolf presence and these variables was not always consistent, even where it is mentioned. We have reread the studies we reviewed and, where altitude, terrain, etc is mentioned, we offer these summaries in support of our choice:

 • The wolf population studied covered a range of elevations, and the higher altitudes (notably a 1100m site) were apparently used as a retreat from humans (Ciucci et al., 1997).

 • North-west exposure and snow permanence were predictors of wolf presence, but not one of the most important factors. Wolves may follow ungulates to lower altitudes in winter, but conversely can also avoid low altitudes because of high human disturbance (Massolo and Meriggi, 1998).

 • In Southern Poland wolf habitat is characterised by higher elevation, but this is not an essential variable for an effective model (possibly because of high correlation with another variable). There seems to be no such relationship observed in Northern Poland. A review of European research shows that wolves are mainly found between 518m -1680m, though this varies by country (Jędrzejewski et al., 2005, 2008).

 • Very high elevations may limit prey availability, though it is not used here as a variable. This potential impact would anyway be accounted for in our paper through our use of a spatial deer density dataset (Mladenoff et al., 1995; Mladenoff and Sickley, 1998).

 • Elevation is included in an Italian study, but only as an ancillary variable highly correlated to human disturbance and cover availability. It’s not subsequently mentioned as being significant (Corsi et al., 1999).

 • Land >1800m had reduced suitability due to severe conditions and lack of prey. However, all of Scotland lies below 1345m, and the vast majority below 1000m, so this is not relevant (Glenz et al., 2001).

 • Wolf use areas in the Northern Rocky Mountain region are characterised by higher elevation and slope than non-use areas. However, core use areas have lower elevation and slope than the entire home ranges, and neither slope nor elevation are found to be a primary factor (Oakleaf et al., 2006).

In the studies where altitude or terrain variables show any correlation with wolf presence, this seem to be due to their covariance with either prey availability or human density, both of which are already represented in our model. Otherwise, they do not emerge as important factors. In contrast, the factors of prey availability, habitat/land cover, human density and road density emerged consistently in the literature as primary factors. We hope this is sufficient justification for the exclusion of terrain or altitude variables from our model.

13) What was the resolution of the Corine Land Cover dataset?

This has now been added (line 327).

14) Two comments asking why human density dataset was not “smudged” as was done for deer.

Though both the deer density dataset and the human density dataset are superficially similar, in that they both represent the density of creatures as spatially static, we feel they are essentially different and do not require the same treatment. Though humans move, human population centres do not (or very rarely/slowly). Therefore, human density at the scale of a country remains relatively stable. However, the deer density dataset represents where deer were found at a certain point in time (at the time of the survey). The areas of high density in the dataset correspond to deer herds, but as these wander the landscape, their realised spatial density is likely to be much less aggregated and fixed. By “smudging” the deer density dataset, we attempt to represent the movement of a herd around its home range, and we do not consider this to be necessary for human density. This is further supported by the fact that the deer density dataset was generally much higher resolution than the human density dataset, i.e. substantial human movement could occur within the large spatial units of the dataset, and the unit density still remain the same.

15) For which habitat is this deer home range size typical?

Sentence amended to read, “As this snapshot did not accurately reflect the realised spatial density of deer over time, kernel density estimation (KDE) was applied to each herd location, with an output cell size of 50m and a search radius of 1480m (representing the average home range of a red deer in Scotland, where they mainly roam open heaths and peatlands)”. Citations for this statement have been added (line 358).

16) Are roads gated? Is access unrestricted?

Sentence amended to read “This dataset includes minor roads but not private roads, and no weighting was applied to roads of different rank, as this does not seem to be common practice in wolf/road studies” (line 382).

17) What was the resolution of the human density dataset?

The spatial units for human density were the census boundaries. These are variable in size, being larger where human density is low, and smaller where it is large. This is stated in the text (line 389).

Discussion

18) Suggestion to change “lochs” to “lakes” or “bogs”.

We agree with this suggestion, it has now been changed to “lakes” (line 483).

19) Comment on the suitability of open heath and bogs for wolves: “If they can walk there and if deer walks there, the the only factor which can prevent wolves from being there are humans.

So far there was not word about livestock (sheep). How many, at that kind of land cover they are kept, what is livestock husbandry practice? I guess sheep and cattle are left grazing unattended. do they stay on pastures over night? Any guarding?

This eventual component of the habitat may prove as more important than anything else, take a look at sheep husbandry and "coexistence" with bears, wolves and wolverines is a country which subsidies rural life, lots of free roaming sheep and very low tolerance toward wolves.”

We are inclined to agree that these open habitats are likely to be suitable for wolves, given their high prey density. We have addressed the remainder of this comment as per point 2 above, and believe all these specifics are now covered.

20) Suggestion to look up “bold wolves”.

We thank the Reviewer for their suggestion. This is an interesting topic, and a useful qualifier to the assumption that high flexibility/tolerance in wolves is always a good thing.

21) Request for further information on the Scottish farming, subsidies, etc.

Please see our response to point 2. We think that the additions to the text outlined there address this comment, too.

From Reviewer 2

General comments

1) From a methodological point of view, I have only a few doubts concerning in particular the suitability scores attributed to the land use classes, especially pastures and open areas without vegetation (Table 2). Pastures can be largely used by wolves in particular when they host several livestock species especially sheep and goats. On the other hand, open spaces without vegetation shouldn’t be suitable for wolves. In fact, being without vegetation they should not be habitats rich in potential prey and, moreover, they do not offer shelter or the possibility of denning. The Authors should justify these choices.

We thank the Reviewer for their comment. We agree that Pastures would technically be suitable for wolves, but we we felt that to include them in proposed suitable habitat for wolves, with an eye to reintroduction, would be to include livestock as an acceptable prey-base. As any reintroduction programme would be seeking to minimise livestock predation, we did not feel this would constitute a useful model, and could lead to a great deal of human-wildlife conflict if it were employed. An extra sentence has been added to the Land Cover section to explain this (line 349). 

 As regards open spaces without vegetation, it is certainly true that in Europe, and to some extent the US, ungulates are mostly associated with forest, and not open habitats. However, Scotland is anomalous, because there is in fact a high density of red deer on its open moorlands and peatlands, where they are maintained by sporting estates (Mitchell et al., 1977; The British Deer Society, 2016; Scottish Natural Heritage, 2018). This was alluded to in our Rationale, but an additional sentence has been added there (line 253), in the Land Cover section (line 343), and in the Discussion (line 540) to make this fact more explicit. The latter sentence also mentions the implications for shelter and denning, as pointed out by the reviewer. However, we also note that wolves have been found to den successfully in chaparral (Kusak et al., 2005) and tundra (McLoughlin et al., 2004), and to be non-habitat-specific in general (Mech, 1995).

Results

2) Four comments pointing out that the main text should illustrate, explain and highlight (and not just describe) what appears in the figures, as this is already stated in the captions.

We thank the Reviewer for their comment. We have revised the text and attempted to differentiate more between the content of the captions, and the content of the main text. We hope the text is now a little more interpretive, and that this concern is addressed (lines 426 – 458).

3) Reverse the order in which the fuzzy membership maps are mentioned in the text to match the order they appear in the figure.

This change has now been applied (line 442).

4) Comment that the sentence “The area and proportion of mainland Scotland falling into each of ten equal classes of suitability was calculated for each model” belongs in Methods.

Sentence has been amended to read “Calculating the area and percentage of mainland Scotland that falls into ten equal classes of suitability (Table 5) shows that the majority of Scotland is unsuitable according to these models, and this does not vary much between models” (line 455). We hope this sentence now belongs in the Results section.

5) Point that Table 5 refers to percentages, not proportions.

Now amended to say “Percentage” (line 463).

Discussion

6) Comment that releasing supplementary animals is impractical, because it leads to conflict with existing packs.

We have been unable to find particular references for this either way. However, we have added a qualifying phrase, so the sentence now reads, “These include the requirements of maintaining wolf genetic diversity and metapopulation, which would require either an area far bigger than that needed to support a few packs, or else regular introductions of additional animals, though this may result in conflict with existing packs” (line 556).

7) Where referring reader to other studies, would be better to summarise the conclusions of those summaries.

We agree and this change has now been applied (line 575-591).

8) A few deletions of unnecessary text throughout.

These changes have now been applied.

Other changes

1) Added mention of Wilson paper to text.

We have now added a reference to Wilson (2004) which reviews attitudes to and potential for reintroduction of large carnivores in the UK. This paper is now referred to in the Introduction (line 95), summarised in the Discussion (line 585), and cited elsewhere.

2) Added mention of Mech paper to text.

Similarly, we have now added a reference to a paper that strikes a cautionary note about the ecological benefits of wolves (Mech, 2012). We felt this should be added to the Introduction in the interests of clarity and balance (line 71).

3) Small editing changes

Some very slight editing changes in aid of grammar or clarity, or to correct formatting, have been made in lines 69, 228, 322 (updating link in table), 324, 353, 379, 383, 386, 454 (caption), 492. None of these alter the content or meaning of the paper.

Bibliography

Ciucci, P., Boitani, L., Francisci, F. and Andreoli, G. (1997) ‘Home range, activity and movements of a wolf pack in central Italy.’ Journal of Zoology, 243(4) pp. 803–819.

Corsi, F., Duprè, E. and Boitani, L. (1999) ‘A Large-Scale Model of Wolf Distribution in Italy for Conservation Planning.’ Conservation Biology. [Wiley, Society for Conservation Biology], 13(1) pp. 150–159.

Glenz, C., Massolo, A., Kuonen, D. and Schlaepfer, R. (2001) ‘A wolf habitat suitability prediction study in Valais (Switzerland).’ Landscape and Urban Planning, 55(1) pp. 55–65.

Jędrzejewski, W., Jędrzejewska, B., Zawadzka, B., Borowik, T., Nowak, S. and Mysłajek, R. W. (2008) ‘Habitat suitability model for Polish wolves based on long-term national census.’ Animal Conservation, 11(5) pp. 377–390.

Jędrzejewski, W., Niedzialkowska, M., Mysłajek, R. W., Nowak, S. and Jędrzejewska, B. (2005) ‘Habitat selection by wolvesCanis lupus in the uplands and mountains of southern Poland.’ Acta Theriologica, 50(3) pp. 417–428.

Kusak, J., Skrbinšek, A. M. and Huber, D. (2005) ‘Home ranges, movements, and activity of wolves (Canis lupus) in the Dalmatian part of Dinarids, Croatia.’ European Journal of Wildlife Research, 51(4) pp. 254–262.

Massolo, A. and Meriggi, A. (1998) ‘Factors affecting habitat occupancy by wolves in northern Apennines (northern Italy): a model of habitat suitability.’ Ecography, 21(2) pp. 97–107.

McLoughlin, P. D., Walton, L. R., Cluff, H. D., Paquet, P. C. and Ramsay, M. A. (2004) ‘Hierarchical Habitat Selection by Tundra Wolves.’ Journal of Mammalogy. Oxford Academic, 85(3) pp. 576–580.

Mech, L. D. (1995) ‘The Challenge and Opportunity of Recovering Wolf Populations.’ Conservation Biology. [Wiley, Society for Conservation Biology], 9(2) pp. 270–278.

Mech, L. D. (2012) ‘Is science in danger of sanctifying the wolf?’ Biological Conservation, 150(1) pp. 143–149.

Mitchell, B., Staines, B. W. and Welch, D. (1977) Ecology of Red Deer: A Research Review Relevant to their Management in Scotland. Institute of Terrestrial Ecology, Natural Environment Research Council.

Mladenoff, D. J. and Sickley, T. A. (1998) ‘Assessing Potential Gray Wolf Restoration in the Northeastern United States: A Spatial Prediction of Favorable Habitat and Potential Population Levels.’ The Journal of Wildlife Management, 62(1) pp. 1–10.

Mladenoff, D. J., Sickley, T. A., Haight, R. G. and Wydeven, A. P. (1995) ‘A Regional Landscape Analysis and Prediction of Favorable Gray Wolf Habitat in the Northern Great Lakes Region.’ Conservation Biology, 9(2) pp. 279–294.

Oakleaf, J. K., Murray, D. L., Oakleaf, J. R., Bangs, E. E., Mack, C. M., Smith, D. W., Fontaine, J. A., Jimenez, M. D., Meier, T. J. and Niemeyer, C. C. (2006) ‘Habitat Selection by Recolonizing Wolves in the Northern Rocky Mountains of the United States.’ The Journal of Wildlife Management, 70(2) pp. 554–563.

Scottish Natural Heritage (2018) ‘Deer Count Density Vector Map.’

The British Deer Society (2016) ‘Deer Distribution Survey 2016.’ The British Deer Society.

Wilson, C. J. (2004) ‘Could we live with reintroduced large carnivores in the UK?’ Mammal Review, 34(3) pp. 211–232.

---

## [Editor Report · Decision Letter 1]

5 Oct 2022

Rule-based habitat suitability modelling for the reintroduction of the grey wolf (*Canis lupus*) in Scotland

PONE-D-22-05836R1

Dear Dr. Gwynn

We’re pleased to inform you that your manuscript has been judged scientifically suitable for publication and will be formally accepted for publication once it meets all outstanding technical requirements.

Kind regards,

Marco Apollonio

Academic Editor

PLOS ONE
---

## [Editor Report · Acceptance letter]

13 Oct 2022

PONE-D-22-05836R1 

Rule-based habitat suitability modelling for the reintroduction of the grey wolf (*Canis lupus*) in Scotland 

Dear Dr. Gwynn:

I'm pleased to inform you that your manuscript has been deemed suitable for publication in PLOS ONE. Congratulations! Your manuscript is now with our production department. 

Kind regards, 

on behalf of

Prof. Marco Apollonio 

Academic Editor

PLOS ONE